# Applying a Random Encounter Model to Estimate the Asiatic Black Bear (*Ursus thibetanus*) Density from Camera Traps in the Hindu Raj Mountains, Pakistan

**DOI:** 10.3390/biology13050341

**Published:** 2024-05-14

**Authors:** Faizan Ahmad, Tomoki Mori, Muhammad Rehan, Luciano Bosso, Muhammad Kabir

**Affiliations:** 1Wildlife Ecology Lab, Department of Forestry & Wildlife Management, The University of Haripur, Khyber Pakhtunkhwa 22600, Pakistan; faizanqandil99@gmail.com (F.A.); mrehanswaat@gmail.com (M.R.); 2Institute for Mountain Science, Shinshu University, Kamiina County, Nagano 380-8544, Japan; tmkmori12@gmail.com; 3Laboratory of Environmental Zoology, Faculty of Agriculture, Meijo University, Nagoya 468-8502, Japan; 4Institute for Agriculture and Forestry Systems in the Mediterranean, National Research Council of Italy, Piazzale E. Fermi, 1, 80055 Portici, Italy; luciano.bosso@cnr.it

**Keywords:** Asiatic black bear, camera traps, conservation, distribution range, random encounter model, vulnerable species

## Abstract

**Simple Summary:**

The Asiatic black bear (*Ursus thibetanus*) is classified as vulnerable according to the Red List of the International Union for Conservation of Nature due to habitat fragmentation and population decline. We used camera traps and a Random Encounter Model (REM) to estimate the population density of Asiatic black bears during the autumn and winter seasons in the Hindu Raj Mountains. We estimated, using the REM, a population density of *U. thibetanus* of 1.875 (standard error = 0.185) per square kilometer, which is significantly higher than that in other habitats. Our results showed that during autumn and winter, the bear population density tends to concentrate at lower elevations. Forest cover showed a positive correlation with the rates of bear encounters unlike the Euclidean distance to human settlements, altitude, and aspect variables. To improve bear conservation, it is very important to determine population density in all seasons at different elevations and to investigate how the depletion or abundance of acorns or other natural food resources is associated with human–bear conflicts in the study area.

**Abstract:**

Estimating the population density of vulnerable species, such as the elusive and nocturnal Asiatic black bear (*Ursus thibetanus*), is essential for wildlife conservation and management. We used camera traps and a Random Encounter Model (REM) to estimate the population density of *U. thibetanus* during the autumn and winter seasons in the Hindu Raj Mountains. We installed 23 camera traps from October to December 2020 and acquired 66 independent pictures of Asiatic black bears over 428 trap nights. Our results showed that the bears preferred lowland areas with the presence of *Quercus* spp. We estimated, using the REM, a population density of *U. thibetanus* of 1.875 (standard error = 0.185) per square kilometer, which is significantly higher than that in other habitats. Our results showed that during autumn and winter, the bear population density tends to concentrate at lower elevations. Forest cover showed a positive correlation with the rates of bear encounters unlike the Euclidean distance to human settlements, altitude, and aspect variables. The approaches used here are cost-effective for estimating the population density of rare and vulnerable species such as *U. thibetanus*, and can be used to estimate their population density in Pakistan. Population density estimation can identify areas where the bears live and human–bear conflicts occurred and use this information in future wildlife management plans.

## 1. Introduction

With the rising intensity and global expansion of anthropogenic effects on habitats and animals, tracking trends in wildlife populations’ abundance and distribution is becoming an increasingly essential conservation goal [1]. Asiatic black bears *(Ursus thibetanus,* referred to hereafter as simply “bears”) are classified as vulnerable according to the IUCN Red List due to habitat fragmentation and population decline [2]. Shiekh [3] showed that about 1000 specimens of bears live in different parts of Pakistan and are mainly threatened by habitat loss, human–wildlife conflicts, food depletion, and poaching [4,5,6]. The most sensitive metric to change is the density of the bear population [1]. Estimating the population density of wild animals helps identify areas of interest for human–wildlife coexistence [7,8]. Hotspots areas where conflicts are prevalent are particularly interesting, from a conservation perspective, due to the interaction between humans and wildlife [9]. The identification of these hotspots may help conservation organizations to prioritize interventions such as implementing deterrent measures or establishing buffer zones to reduce conflicts and protect both people and wildlife [10]. Conflicts are more likely to occur when human settlements are built into wildlife areas, when agriculture is practiced, and when natural resources are scarce [11]. Comprehending this information and integrating it into conservation plans facilitates the development of focused initiatives to mitigate conflicts and reduce their consequences through the implementation of measures such as creating wildlife corridors or protected areas [7,12,13].

An accurate and reproducible estimation of population density is crucial for managing and conserving threatened species, particularly those facing the risk of extinction [14,15]. However, obtaining precise and reliable population density estimates can be challenging due to cost constraints and the need for accurate and feasible methods for management purposes [16]. To overcome these challenges, researchers often use camera traps as effective tools for gathering reliable quantitative data on elusive or nocturnal species, as well as those that are solitary or exist at low densities [17,18]. In recent years, the use of camera traps has gained popularity as a cost-effective and non-invasive method for estimating the population density of wildlife species in a wide range of habitats, proving to be particularly efficient [19,20].

Promising approaches for estimating population density from camera-trap data are the Spatially Capture–Recapture (SCR) and the Open Population (OP) methods, which combine spatial and temporal information from photographs of individually recognizable animals [1,21,22,23,24]. SCR and OPSCR stand as the only methods possessing enough power to consistently identify moderate to significant (from 20% to 80%) decreases in population density [1]. SCR analysis has been widely applied to spotted and striped felids, but most wildlife species do not have natural marks that enable individual recognition, limiting the applicability of Capture–Recapture methods that require the physical capture and tagging of animals [25,26,27].

To address this limitation, Rowcliffe et al. [28] developed the Random Encounter Model (REM) as a method for estimating population density without the need for individual recognition. The REM is based on modeling random encounters between moving animals and stationary camera traps, considering key variables that affect the encounter rate (i.e., the number of animals detected per sampling unit), such as the camera detection zone (defined by its radius and angle) and the daily distance traveled by the animals (i.e., day range). The advantage of the REM is that it does not require individual identification, making it suitable for monitoring both marked and unmarked populations, without the need for animal capture and tagging [28,29]. Since its development, the REM has been widely used to estimate the population density of unmarked populations and has been recommended as a reliable method compared to other approaches [30,31]. Several REM studies have compared REM-derived density estimates with reference densities for various species, including gregarious and non-gregarious carnivores, ungulates, and other wildlife [30,32,33,34].

In this study, we investigated how the seasonal migration of bears to lower elevations during autumn and winter influences their population density, spatial distribution, and interactions with human settlements in the Hindu Raj Mountains. Furthermore, we analyzed our results from the perspective of the implications for bear conservation and management in this ecosystem. We used the REM to estimate the population density of bears in the moist temperate zone of the Hindu Raj Mountain range, Pakistan. This study is important because the population density of bears in the study area still remains undocumented and this information can be used to implement measures to mitigate human–bear conflict.

## 2. Materials and Methods

### 2.1. Study Area

We conducted this study in an area that extends for 739 km^2^, between 35°0′20″ and 35°28′10″ N latitudes and 72°22′45″ and 72°48′15″ east longitudes. This region features moist temperate, dry temperate, subalpine, alpine, and snowcap zones between 1,235 and 5,954 m above sea level (a.s.l.) (Figure 1). The Hindu Raj Mountain range is situated between the Himalayan and Hindukush Mountain ranges [35]. The average annual temperature is 10.8 °C, with a high of 32.1 °C and a low of −12.2 °C; the yearly rainfall is 1029 mm [36]. In addition to bears, several other species live in this area, such as the snow leopard (*Panthera uncia*), the flare-horned markhor (*Capra falconeri falconeri*), the leopard cat (*Prionailurus bengalensis*), the yellow-throated marten (*Martes flavigula*), the golden jackal (*Canis aureus*), the red fox (*Vulpes vulpes*), the gray wolf (*Canis lupus*), the rhesus monkey (*Macaca mullata*), the Indian crested porcupine (*Hystrix indica*), and the giant Indian flying squirrel (*Petaurista petaurista*) [9,37].

### 2.2. Data Collection

We utilized the grid index feature in ArcGIS version 10.8 to divide the study area into 2 × 2 km grids [38]. In each grid, we selected random plots and placed cameras at random distances from a starting point on the grid line. Randomization of the camera stations is required for the REM [39]. We installed a total of 23 camera traps (ZopuCam, SL122C-2, Shenzhen Zhuopu Digital Technology Co., Ltd. Guangdong, China, https://www.zopudt.com/ (accessed on 15 October 2023)), facing north, at least one kilometer apart from each other (from October 2020 to December 2020) for a trapping duration of 428 nights, covering deep-snow-free zones of the forest. Usually these are suitable areas for bears, as they avoid deep snow and open alpine meadows at high altitudes and move to lower altitudes during autumn and winter [9]. To capture all animal movement, we set the camera traps to trigger three rapid-fire photos, with minimal delay between each trigger. In fact, the recording of successive photo series was triggered one second after the previous triggering. Our camera traps utilized an infrared flash to capture nocturnal photos, ensuring comprehensive coverage throughout the 24 h period [29]. To ensure independent detections for calculating activity levels, we deleted from our dataset multiple detections obtained from the same camera-trap station within 30 min [40]. The camera traps were not baited. The final sampling sites differed from the original design in certain grids to accommodate the placement of camera traps on trees and bypass particularly challenging terrain (such as steep cliffs). Throughout the setup of all camera traps, natural markers like big stones and branches were positioned at predetermined distances within the camera-trap stations. These markers were subsequently utilized to determine the locations and distances covered by the bears detected with the camera traps.

### 2.3. Data Analysis

The REM utilizes a technique that relies on random interactions between the animals and cameras, taking into account all factors that influence the rate of encounters [28]. The formula used to estimate the population density from the encounter rate and other related parameters is as follows:D = Y/H × π/v × r × (2 + θ)(1)

In this equation, Y represents the total number of encounters, which is defined as the number of independent picture sequences captured via the camera trap; H denotes the total camera survey effort, which is the combined operational time of all camera traps; *v* is the average distance an individual animal travels in a day (24 h), also known as the day range; and the parameters r and θ represent the radius and angle of the camera trap’s detection zone, respectively. An event was classified as independent when an individual moved into and then out of the camera trap’s field of view. The day range (*v*) was calculated as the product of the animal’s speed and its level of activity. The speed, denoted as *si*, was calculated by dividing the distance *di* (m) by the time *ti* (sec). The time *ti* represented the duration between the first and last photo of the capture event, accounting for the movement of the bear. First, speed was measured for each sequence by dividing the distance traveled by the duration of the sequence. Subsequently, the bear activity level was estimated following the method described by Rowcliffe et al. [41] by using the R package activity [42,43] (Figure 2; Appendix A). Sequences in which animals responded to the camera trap were included in the encounter rate calculation but were excluded from the speed analysis [44]. Finally, we estimated the daily range following the procedure described by Palencia et al. [34]. The method used for estimating the daily distance travelled had a large impact on density estimates from the REM. We used estimates of the day range from the camera traps in line with the studies by Rowcliffe et al. [44] and Palencia et al. [34], which demonstrated that camera-trap-based day range estimates were 1.9 to 7.3 times greater than those derived from telemetry data. They argued that the distance traveled could be underestimated in tracking data where spatial locations are not recorded frequently enough to capture small-scale movements. The day range was calculated as the product of the average speed and the activity level. We documented the position (radial distance and angle) of the animal when it initially activated the camera trap. The levels of variance associated with the encounter rate, detection radius (mean), detection angle (mean), and speed were also estimated, applying the bootstrap method with 10,000 iterations using the “boot” and “readr” packages in R statistical software Vers. 4.3.3 (further details can be found in the Appendix A) [42]. The total variance in the density estimates was calculated using the delta method, which can consider the variances of all parameters, including the encounter rate, the day range, and the radius and angle of detection (further details are provided in the Appendix A) [45,46].

### 2.4. Hotspot Analysis

To identify the hotspots of bear encounters within the study area, we employed methods based on Kernel Density Estimation (KDE) [47,48]. KDE is a widely used technique for visualizing and analyzing spatial data, with the objective of understanding and potentially predicting patterns of events. Utilizing the “Kernel Density for Point Features” tool in ArcGIS 10.8, we calculated the point feature density around each output raster cell. The algorithm of this tool fits a smoothly curved surface over each point, with the surface value being highest at the point location and decreasing as the distance from the point increases, eventually reaching zero at the search radius (bandwidth) [49]. This approach is particularly effective in identifying hotspots because it makes a series of density estimates over a grid that covers the entire point pattern [50]. By applying KDE to our bear encounter data, we were able to identify areas of high bear activity visually and quantitatively, which are crucial for understanding spatial patterns and informing conservation efforts.

### 2.5. Exploratory Regression Analysis

We employed the exploratory regression tool in ArcGIS 10.8 to investigate the connection between bear encounter rates and a set of environmental variables such as the Euclidean distance to human settlements, altitude, forest cover, aspect, slope, and roughness (further details in the Appendix A) [38,51]. To investigate the connection between the bear encounter rates and environmental variables, we utilized raster data for altitude, aspect, slope, and roughness, which were downloaded from Open Topography [52]. Forest-cover data were obtained from the Global Forest Change Project [53]. The Euclidean distance to human settlements was determined using a raster developed from Esri’s land-use land-cover data in ArcGIS 10.8 [54]. These environmental variables were selected based on their potential influence on the bears’ habitat preferences and movement patterns. The analysis systematically evaluated all possible combinations of the input explanatory variables, aiming to identify Ordinary Least-Squares models that best explained the dependent variable, i.e., the encounter rates, within user-specified criteria [55,56]. Key parameters set for the analysis included a maximum number of explanatory variables of 6, a minimum acceptable adjusted R-squared value of 0.3, a maximum coefficient *p*-value cutoff of 0.05, a maximum Variance Inflation Factor (VIF) value cutoff of 7.5, and a minimum acceptable *p*-value for the Jarque–Bera and spatial autocorrelation tests of 0.1. The results from this analysis provided insights into the ecological dynamics influencing the bear distribution, serving as a robust framework for understanding habitat preferences and guiding conservation efforts.

## 3. Results

### 3.1. Bear Density

The camera traps acquired 66 independent detections of bears during a survey effort realized in 428 trap nights. The overall encounter rate value was 0.143 ± 0.031 individuals (cam × day)^−1^ (mean ± SE). The encounter rate’s variance, assessed using the bootstrap technique with 10,000 iterations, was 0.001. The encounter rate was highest for the altitude range of 2000–2500 m (1.74 ± 0.06, mean ± SE), followed by 1500–2000 m (0.8 ± 0.08, mean ± SE) and 2500–3000 m (0.78 ± 0.03, mean ± SE; Figure 2 and Figure 3).

We summarize the estimated REM parameters in Table 1. The activity level calculated for the bear population had a mean of 0.41 (0.27–0.55), with an SE of 0.071 (Figure 4). The day range for the bear population averaged at 21.39 ± 4.33 km/day (mean ± SE).

The average radius and angle of detection values were about 0.0045 ± 0.00025 km (mean ± SE) and 0.426 ± 0.029 radians (mean ± SE), respectively.

### 3.2. Hotspots of Bear Encounter Rates

Based on the KDE analysis conducted within our study area, we showed the hotspot areas (in red) with higher rates of bear encounters than other areas (green), demonstrating a pronounced concentration of bear activity at lower altitudes near human settlements. The gradient of colors, from green to red, effectively visualizes the increasing point feature density, with the most intense red areas indicating the highest frequency of bear encounters. This spatial pattern underscores the influence of the altitude and proximity to human developments as key factors in bear distribution during the autumn and winter seasons (Figure 5).

### 3.3. Results of Exploratory Regression Analysis

Regression analysis showed that the correlation between bear encounter rates and various environmental variables show on one hand that the altitude was the most important variable (100%) and had a negative link with the encounter rates, such as the Euclidean distance to human settlements (importance variable: 71.88%) and the aspect (importance variable: 3.12%). On the other hand, forest cover was an important variable (importance variable: 40.62%) and had a positive correlation with the encounter rates. Slope and roughness were not significant (further details in the Appendix A).

The best models were selected based on the highest adjusted R-squared values. We used the final model with the highest adjusted R-squared value (0.66) including the Euclidean distance to human settlements, altitude, forest cover, and aspect as predictors. All of the chosen models passed the multicollinearity test with a maximum VIF of less than 7.50. Most models passed the Jarque–Bera test for residual normality as well as the spatial autocorrelation test.

## 4. Discussion

Here we present the results of the population density estimation for the bear in autumn and winter by using a REM approach. Our results showed that the estimated population density of the bears in our study area was 1.875 (SE = 0.185) per square kilometer. Interestingly, our value is significantly higher than the population density values reported for this species in previous studies conducted in different regions. For example, a study conducted in Shirakawa Village, Gifu Prefecture, Japan reported a population density of 55 individuals per 100 km^2^ [57], while a study conducted at the Daranghati Wildlife Sanctuary (WLS) and Rupi Bhaba WLS in India reported a population density of 2.5 individuals/100 km^2^ [58]. Earlier studies from India estimated the density of the bears as 130 to 180 per 100 km^2^ through questionnaire surveys and genetic analyses of collected hair samples, which were available only from Dachigram National Park, Jammu, and Kashmir [59,60,61]. Similarly, a study from the Senchal WLS in West Bengal estimated the tentative density of the species to be 28.57/100 km^2^ [62]. Ngoprasert et al. [63] estimated the population density of bears in the evergreen forests of Thailand to be 8 ± 3.04 individuals/100 km^2^ and 5.8 ± 2.31 individuals/100 km^2^ using SCR models with unique chest marks, respectively. The estimated minimum density of the bears in the Annapurna Conservation Area, Nepal was 11.5 bears/100 km^2^, while the estimate from the National Parks of the western Himalayas was 17 bears/100 km^2^ [64,65]. Furthermore, a study in Russia estimated the black bear density to be 8–11 bears/100 km^2^ [66].

We observed higher bear encounter rates at lower elevations, which we attribute to the fact that this study was conducted in autumn and winter, when bears migrate from habitats in higher elevations with deep snow and limited food resources to lower elevations with a greater availability of food such as acorns. This finding aligns with previous research that indicated that bears move in response to food availability [9,60]. The abundance of the local bear population can be boosted by supplemented feeding and a plentiful natural food supply [2,9,67,68,69,70,71,72]. In our study area, abundant food resources, such as acorns or maize and *Diospyros lotus* fruits in fields and orchards are used by the bears as supplementary food at lower elevations [9].

Conflicts between humans and bears arise from the higher frequency of bear encounters close to orchards, agricultural areas, and human settlements. Our exploratory regression analysis revealed a negative relationship between bear encounter rates and the bears’ distance from human settlements, indicating that encounter rates decrease as this distance increases in autumn and winter seasons. Additionally, our analysis showed a positive relationship between bear encounter rates and the forest cover. Bears are particularly associated with oak forests at lower elevations, which provide a crucial food source in the form of acorns during hyperphagia, especially in autumn and winter [9,73]. However, the local communities’ harvesting of acorns, wood, and leaves from *Quercus* species increases the risk of human–bear conflicts by depleting natural resources and forcing bears to rely more on human-related food [9,63]. Therefore, a shortage of natural food resources can result in a major decline in the black bear population because of retaliatory killings [74,75].

Therefore, considering all factors, the overall population density of bears in the whole study area may not be much higher than that reported in other studies in different regions, and it is important to consider the seasonal concentration of bears in a certain region when calculating population density estimates [57,58,59,61,62,63,64,65,68,76,77,78,79].

This study showed autumn and winter population density estimates for the Asiatic black bear in northern areas of Pakistan combining camera-trap and REM methods. The estimated density was significantly higher than that reported in previous studies conducted in other regions. It is important to consider the seasonal concentration of bears in a certain region when calculating population density estimates. The migration of bears from higher elevations in search of food throughout the autumn and winter seasons was linked to their high population density at lower elevations in the research area. This migration also frequently leads to the interaction between humans and bears. The availability of abundant food resources at lower elevations, such as acorns in the wild and maize and *D. lotus* fruits in fields and orchards, attracts bears to these areas. The significance of managing habitats at lower altitudes should be highlighted to address the increased density of local bear populations during the winter season. This could involve protecting and enhancing suitable habitats, ensuring seamless connectivity between them, and reducing disturbances in these areas. Further research on these bears in the study area could be of interest to analyze, for example, the occurrence of ecological corridors or the impact of climate change on the bears’ distribution and activity [80,81]. To improve bear conservation, it is very important to determine population density in all seasons at different elevations and to investigate how the depletion or abundance of acorns or other natural food resources is associated with human–bear conflicts in the study area.

## Figures and Tables

**Figure 1 biology-13-00341-f001:**
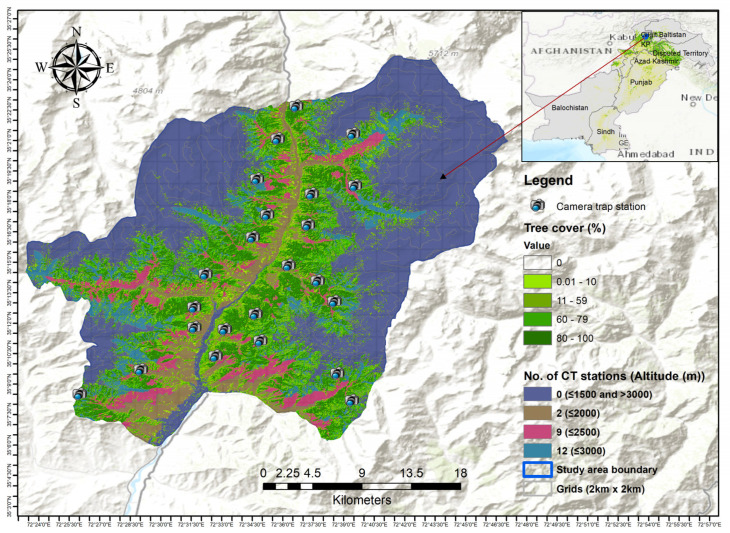
Map of the study area and camera traps.

**Figure 2 biology-13-00341-f002:**
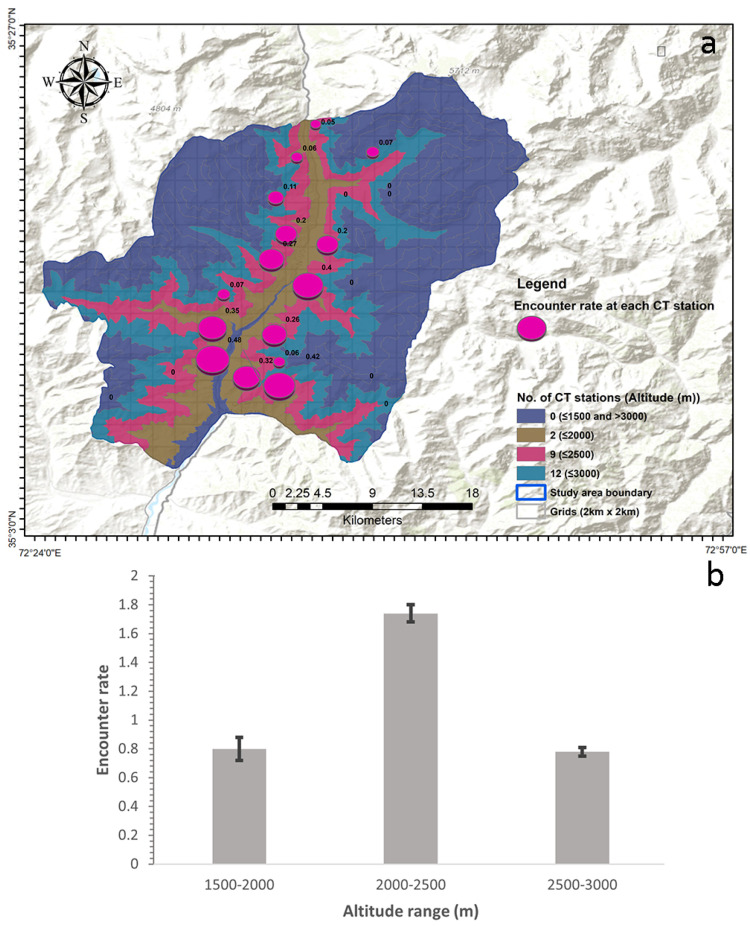
Map showing the encounter rate at each camera-trap station distributed across different altitude gradients (**a**) and the encounter rate in each altitude range of the study area (**b**).

**Figure 3 biology-13-00341-f003:**
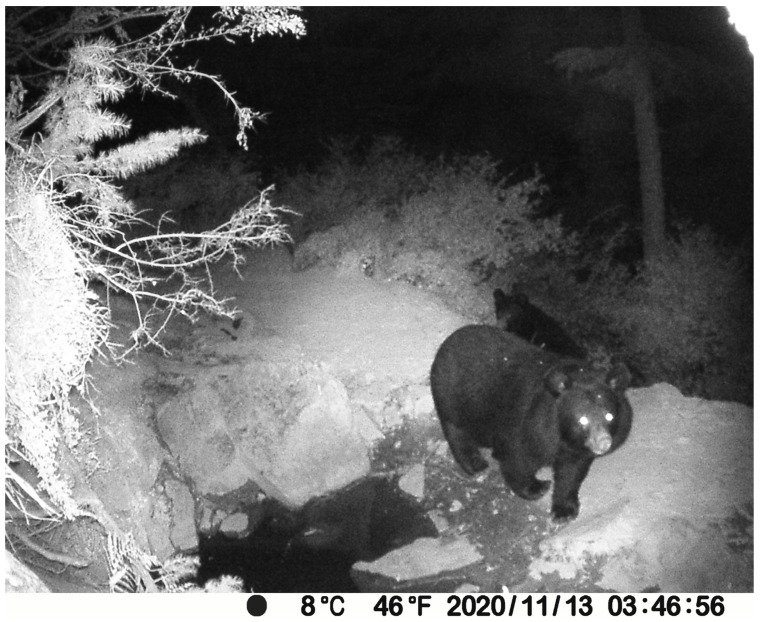
Adult specimen of the Asiatic black bear species captured by means of a camera trap in the study area.

**Figure 4 biology-13-00341-f004:**
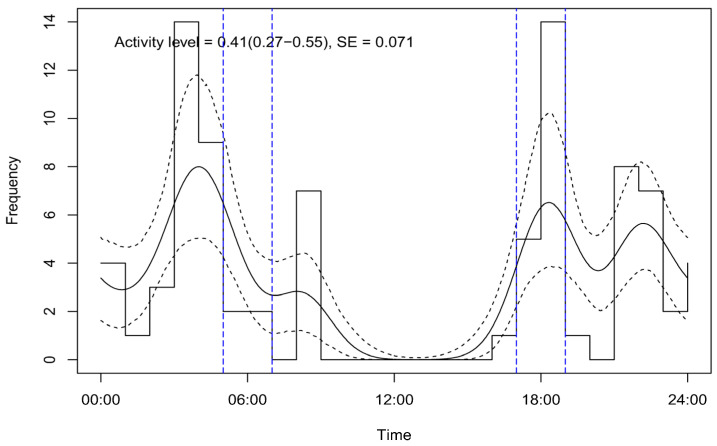
Asiatic black bear activities depicted using the data distribution and fitted circular Kernel Density model. The curved line illustrates diel activity patterns of the species, and the dotted curve lines reflect the lower and upper confidence intervals. The area between the straight lines shows the approximate sunrise and sunset times during our surveyed period of camera trapping.

**Figure 5 biology-13-00341-f005:**
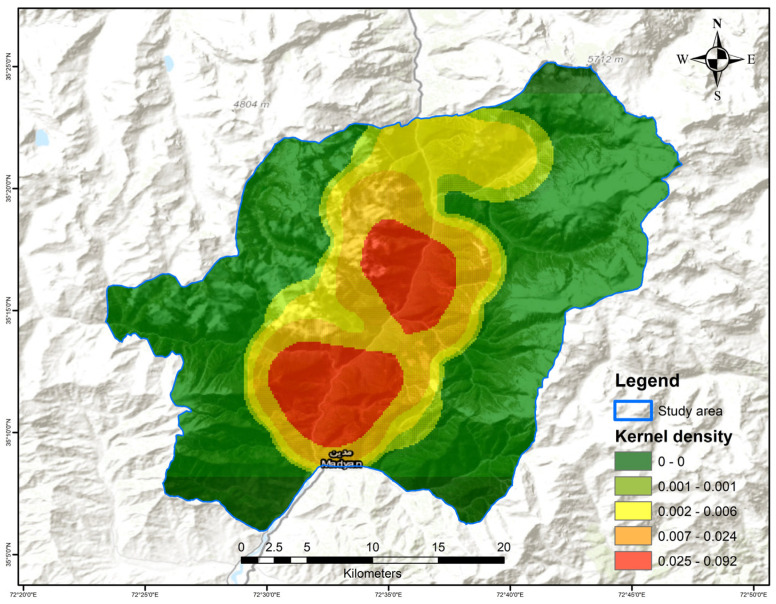
Kernel Density hotspots for bear encounter rates in the study area. Areas with high encounter rates are depicted using a color gradient from green (low density) to red (high density), with the red zones representing the most significant hotspots.

**Table 1 biology-13-00341-t001:** Estimated Random Encounter Model (REM) parameter values for the Asiatic black bear population, where Y/H is the encounter rate; *v* is the average distance travelled by an individual (ind.) bear during the day (day range); r is the radius of detection; and θ is the angle of detection.

Species	Y/H (ind.·(cam·day)^−1^)	*v* (km day)^−1^	r (km)	θ (rad)	Density (N.ind./km^2^)	SE
Asiatic black bear	0.143	21.39	0.0045	0.426	1.875	0.185

## Data Availability

All of the data are available in the article and the Appendix A.

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
