# Peer review of "Applying a Random Encounter Model to Estimate the Asiatic Black Bear (Ursus thibetanus) Density from Camera Traps in the Hindu Raj Mountains, Pakistan"

_biology, 2024, doi:10.3390/biology13050341_

Round 1
Reviewer 1 Report
Comments and Suggestions for Authors
The is an interesting paper on the application of the Random Encounter Model to estimate the population densities of Asiatic black bears. It is great to see the usefulness of the method demonstrated for species of conservation concern. However, it seems that the approach used here underestimates the uncertainty of the popaultion density estimates. This issue needs to be adressed before the article can be considered for publication.
Specific comments
Line 15/26: It does not make sense to write “We used a camera trapping-based method and the Random Encounter Model” since the REM is a camera trap-based method. It would better to write: “we used camera trapping and and the Random Encounter Model […]”
Line 20/33: Please use “concentrate” instead of squeeze.
Line 24: I would rather write: "Population density estimates are important information [...]".
Line 25: What exactly are you referring to with “specific behavioral and ecological characteristics”? Please be more precise.
Line 25: Please use "their" instead of "its" as "bears" is plural.
Line 34: I would suggest to write “our approach is a cost-effective method”.
Line 45: Please write “species” instead of “bear”.
Line 71: Please use “combine” instead of “combines”
Line 74: The problem is that REM is included here, so this structure of the introduction does not make much sense. The way you structured the introduction it seems like REM was created to improve upon other methods with unmarked animals and that is simply misleading. Instead, I would reword this paragraph to just say that SECR is an established reliable framework and follow that up with the last two sentences of the paragraph.
Line 94: Please write “The population density of bears […]”
Line 95: I would suggest to write “The population density of bears in the study area was previously unknown.”
Line 108: Please write “bears” instead of “the bear”.
Line 125: Please write “move” instead of “moves”.
Line 128: Please write “The recording of the next photo series could be triggered […]”.
Line 127: Didn't you define the events based on the 30-minute interval as you wrote previously instead of individuals entering and leaving the field of view?
150: How did you measure distance travelled in the field? Did you describe a measurement tape? Please describe the process in more detail.
Line 160: Sentence is duplicated (line 149).
Line 161: Please specify how you calculated the effective detection distance and angle? Was it the mean of the observed distances and angles?
Line 165: The variances are incorrectly considered, based on what I saw in the Supplementary Material. You cannot just multiply the standard errors of two parameters to obtain the total standard error. You would either need to apply the delta method (see e.g. Powell et al. (2007) ’Approximating variance of demographic parameters using the delta method: a reference for avian biologists’) or – what I would rather suggest - to incorporate the other model parameters in the non-parametric bootstrap. For this, you can resample camera trap locations with their observation events and deployment days and you can resample events for the estimation of the detection radius, angle and day range.
Line 171: Please write “bears” instead of “the bear”.
Line 185: Please write “We summarized the estimated REM parameters in Table 1”.
Line 186: Please write “had a mean of”.
Line 197: Please write “lower and upper confidence intervals”.
Line 205: The word combination does not make sense here. The REM is based on camera trap data!
Line 217: The word “instead” does not make sense in this context.
Line 217: Please write “bears” instead of “the bear”.
Line 225: I would suggest write: “[…] migrate from habitats in higher elevations with deep snow and limited food resources to lower elevations with abundant acorns. This leads to a local concentration of bear individuals.”.
Line 236: Please write “close to human settlements”.
Line 244: There is a single “t” in the text.
Line 252: The different food sources have been discussed before already. Please do not repeat this information here.
Line 257: Please reword, e.g. “Further research on the bears in the study areas could be of interest to analyse for example the occurrence of ecological corridors […].”
Line 259: Please remove “the” before “climate change”.
Best wishes for the revision!
Comments on the Quality of English LanguagePlease see above for my suggestions to improve the writing.
Author Response
The is an interesting paper on the application of the Random Encounter Model to estimate the population densities of Asiatic black bears. It is great to see the usefulness of the method demonstrated for species of conservation concern. However, it seems that the approach used here underestimates the uncertainty of the population density estimates. This issue needs to be addressed before the article can be considered for publication.
Response: We appreciate the reviewer's concern regarding the potential underestimation of uncertainty in our population density estimate. To address this, we have implemented a two-step approach to ensure a comprehensive assessment of uncertainty. First, we employed bootstrapping with 10,000 iterations to estimate the variance associated with all parameters involved in the Random Encounter Model, including the encounter rate, day range, and the radius and angle of detection. This bootstrapping approach allowed us to capture the variability in each parameter, providing a robust estimation of their respective variances. Second, we used the delta method to calculate the total variance in density estimates, considering the variances of all parameters obtained from the bootstrapping step. By integrating the variances of all relevant parameters, the delta method provides a more accurate and conservative estimation of the total variance in density estimates. We have added a detailed description of this approach and its results to the manuscript in (Supplementary materials appendix 1 and 2), demonstrating its effectiveness in providing a more realistic assessment of uncertainty.
Specific comments
Line 15/26: It does not make sense to write “We used a camera trapping-based method and the Random Encounter Model” since the REM is a camera trap-based method. It would better to write: “we used camera trapping and the Random Encounter Model […]”
Response: Thank you for the suggestion. We've revised the sentence to "We used camera trapping and the Random Encounter Model [...]" for clarity.
Line 20/33: Please use “concentrate” instead of squeeze.
Response: Thank you for the recommendation. We've replaced "squeeze" with "concentrate" to enhance readability.
Line 24: I would rather write: "Population density estimates are important information [...]".
Response: Thank you for the suggestion. We've revised line 24 to read: "Population density estimates are important information [...]"
Line 25: What exactly are you referring to with “specific behavioural and ecological characteristics”? Please be more precise.
Response: We have replaced "specific behavioral and ecological characteristics" with a detailed description of the Asiatic black bear's traits, including its elusive nature, nocturnal habits, preference for difficult terrain, arboreal behavior, seasonal patterns such as hibernation, vulnerability to threats, and complex social dynamics, to provide a more precise understanding of the challenges in studying this species. Thank you!
Line 25: Please use "their" instead of "its" as "bears" is plural.
Response: Thank you for the correction. We've replaced "its" with "their" to correctly reflect the plural form of "bears."
Line 34: I would suggest to write “our approach is a cost-effective method”.
Response: Thank you for the suggestion. We've revised the sentence as per reviewer 3 suggestions.
Line 45: Please write “species” instead of “bear”.
Response: Thank you for the suggestion. We've revised the sentence as per reviewer 3 suggestions.
Line 71: Please use “combine” instead of “combines”.
Response: Done as suggested. Thank you!
Line 74: The problem is that REM is included here, so this structure of the introduction does not make much sense. The way you structured the introduction it seems like REM was created to improve upon other methods with unmarked animals and that is simply misleading. Instead, I would reword this paragraph to just say that SECR is an established reliable framework and follow that up with the last two sentences of the paragraph.
Response: Thank you for pointing out the issue with the structure of the introduction. We've removed the reference to REM in line 74 and reworded the paragraph to emphasize that SECR is an established reliable framework, followed by the last two sentences as suggested.
Line 94: Please write “The population density of bears […]”
Response: Done as suggested. Thank you!
Line 95: I would suggest to write “The population density of bears in the study area was previously unknown.”
Response: Done as suggested. Thank you!
Line 108: Please write “bears” instead of “the bear”.
Response: Done as suggested. Thank you!
Line 125: Please write “move” instead of “moves”.
Response: Done as suggested. Thank you!
Line 128: Please write “The recording of the next photo series could be triggered […]”.
Response: Done as suggested. Thank you!
Line 127: Didn't you define the events based on the 30-minute interval as you wrote previously instead of individuals entering and leaving the field of view?
Response: The 30-minute interval was only used for events from which we calculated activity pattern and activity level as suggested by Ridout and Linkie, (2009) (Estimating Overlap of Daily Activity Patterns from Camera Trap Data) to exclude these photos from dataset because of concerns about lack of independence. For other parameters an event was classified as independent when an individual moved into and then out of the camera trap's field of view.
150: How did you measure distance travelled in the field? Did you describe a measurement tape? Please describe the process in more detail.
Response: The day range (v) was calculated as the product of the animal's speed and its level of activity. The speed, denoted as si​, is calculated by dividing the distance di​ by the time ti​. The distance di​ is the estimated distance (in meters) that bear walked during a capture event, as observed from the sequence of photos. This distance is determined by visually estimating the difference between the animal's positions in the first and last photos of the event. The time ti​ is the duration (in seconds) between the first and last photo of the capture event, accounting for the movement of the bear. First, speed was measured for each sequence by dividing the distance travelled by the duration of the sequence.
Line 160: Sentence is duplicated (line 149).
Response: Duplicated sentence deleted. Thank you!
Line 161: Please specify how you calculated the effective detection distance and angle? Was it the mean of the observed distances and angles?
Response: Yes, it was mean of the observed distances and angles.
Line 165: The variances are incorrectly considered, based on what I saw in the Supplementary Material. You cannot just multiply the standard errors of two parameters to obtain the total standard error. You would either need to apply the delta method (see e.g. Powell et al. (2007) ’Approximating variance of demographic parameters using the delta method: a reference for avian biologists’) or – what I would rather suggest - to incorporate the other model parameters in the non-parametric bootstrap. For this, you can resample camera trap locations with their observation events and deployment days, and you can resample events for the estimation of the detection radius, angle and day range.
Response: Done as suggested (Supplementary materials appendix 1 and 2). Thank you!
Line 171: Please write “bears” instead of “the bear”.
Response: Done as suggested. Thank you!
Line 185: Please write “We summarized the estimated REM parameters in Table 1”.
Response: Done as suggested. Thank you!
Line 186: Please write “had a mean of”.
Response: Done as suggested. Thank you!
Line 197: Please write “lower and upper confidence intervals”.
Response: Done as suggested. Thank you!
Line 205: The word combination does not make sense here. The REM is based on camera trap data!
Response: The word combination has been changed. Thank you!
Line 217: The word “instead” does not make sense in this context.
Response: Done as suggested. Thank you!
Line 217: Please write “bears” instead of “the bear”.
Response: Done as suggested. Thank you!
Line 225: I would suggest write: “[…] migrate from habitats in higher elevations with deep snow and limited food resources to lower elevations with abundant acorns. This leads to a local concentration of bear individuals.”.
Response: Thank you for the suggestion. We've revised line 225 to read: "[...] migrate from habitats in higher elevations with deep snow and limited food resources to lower elevations with abundant acorns. This leads to a local concentration of bear individuals."
Line 236: Please write “close to human settlements”.
Response: Done as suggested. Thank you!
Line 244: There is a single “t” in the text.
Response: Deleted the single “t”. Thank you!
Line 252: The different food sources have been discussed before already. Please do not repeat this information here.
Response: Deleted the repeated information. Thank you!
Line 257: Please reword, e.g. “Further research on the bears in the study areas could be of interest to analyse for example the occurrence of ecological corridors […].”
Response: Done as suggested. Thank you!
Line 259: Please remove “the” before “climate change”.
Response: Done as suggested. Thank you!
Reviewer 2 Report
Comments and Suggestions for Authors
Please refer to the pdf document for detailed comments.

Please refer to the pdf document for detailed comments.
Author Response
Estimating the autumn and winter population density of Asiatic Black Bears (Ursus thibetanus) in the Hindu Raj Mountains by using camera trapping and random encounter model,
Faizan Ahmad , Tomoki Mori , Muhammad Rehan , Luciano Bosso , Muhammad Kabir
Title: Estimating the autumn and winter population density of Asiatic Black Bears (Ursus thibetanus) in the Hindu Raj Mountains by using camera trapping and random encounter model
Suggest change to “Estimating the autumn and winter population densities of Asiatic Black Bears (Ursus thibetanus) in the Hindu Raj Mountains by using camera traps and a random encounter model” Two densities are stated and “camera traps” is better English.
Response: We calculated a combined density for both seasons as we didn’t have enough independent detections for each season. In Rovero et al. (2013) there is a personal communication from Rowcliffe that suggests the minimum number of photographs should be at least 50. Other suggestions have been incorporated. Thank you!
Simple Summary: “The Asiatic black bear (Ursus thibetanus) has been classified as vulnerable by the Red List of the International Union for Conservation of Nature, due to its fragmented habitat, and a clear decrease in population. We used a camera trapping-based method and the Random Encounter Model “
Suggest change to: “The Asiatic black bear (Ursus thibetanus) is classified as vulnerable by the Red List of the International Union for Conservation of Nature due to its fragmented habitat and a clear decrease in population. We used camera traps and the Random Encounter Model” Simpler and
better English.
Response: Changes made as suggested. Thank you!
Line 17: “The estimated population density of U. thibetanus by REM was 1.93 (Standard Error = 0.02) per square kilometer, which was significantly higher than the estimated densities of U. thibetanus in other habitats.”
Suggest change to: “The U. thibetanus population density estimated by REM was 1.93 (Standard Error = 0.02) per square kilometre, significantly higher than in other habitats.” More succinct and better grammar and spelling.
Response: Changes made as suggested. Thank you!
Line 19: “The study found that during autumn and winter, bear density tends to squeeze at lower elevations.
Suggest change to: “The study found bear density tends to decrease at lower elevations during autumn and winter.” Simpler English. I do not understand what the author means here by “squeeze”. If it is decrease it is better to say that or increase if that is what is meant.
Response: We have replaced the word “squeeze” with “concentrate” as suggested by Reviewer 1. Other Changes made as suggested. Thank you!
Line 25: “Specific behavioral and ecological characteristics and its presence in inaccessible areas, make the Asiatic black bears (Ursus thibetanus) a difficult species to study.”
Suggest change to: “Specific behavioural and ecological characteristics and its presence in inaccessible areas make the Asiatic black bear (Ursus thibetanus) difficult to study.” Simpler and British spelling. I assume that by using metric measures the authors prefer British spelling.
Response: We have revised this sentence as per suggestions of Reviewer 1. Thank you!
Line 26: “We used a camera trapping-based method and
Suggest change to: “We used camera traps and...” More succinct.
Response: We have revised this sentence. Thank you!
Line 28: “From October to December 2020, we installed 23 camera traps acquiring 66 independent pictures of U. thibetanus during a survey effort of 428 trap nights.
Suggest change to: “By installing 23 camera traps from October to December 2020 we acquired 66 independent pictures of U. thibetanus over 428 trap nights.” Simpler English.
Response: Changes made as suggested. Thank you!
Line 31: “The estimated population density of U. thibetanus by REM was 1.93 (Standard Error = 0.02) per square kilometer, which was significantly higher than the estimated densities of U. thibetanus in other habitats.”
Suggest change to: “The population density of U. thibetanus estimated by REM was 1.93 (Standard Error = 0.02) per square kilometre, significantly higher than U. thibetanus densities estimated in other habitats.” Simpler English.
Response: Changes made as suggested. Thank you!
Line 33: “The study found that during autumn and winter, bear density tends to squeeze at lower elevations. See above for Line 19.
Response: We have revised this sentence. Thank you!
Line 34: “The approaches here used are cost-effective for the estimation of population density for rare and vulnerable species such as U. thibetanus and can be used to estimate its population density in Pakistan. Population density estimation can help in identifying conflict hotspots and mitigating human-bear conflict.”
Suggest change to: “The approaches used here are cost-effective for estimating population density for rare and vulnerable species such as U. thibetanus and can be used to estimate its population density in Pakistan. Population density estimation can identify conflict hotspots and mitigate human- bear conflict.” Simpler English.
Response: Changes made as suggested. Thank you!
Line 42: “The Asiatic black bear (Ursus thibetanus, referred to hereafter as bears) has been classified as vulnerable by the Red List of the International Union for Conservation of Nature, due to its fragmented habitat, and a clear decrease in population [2]. According to [3] approximately 1,000 individuals of the bear exist in different parts of Pakistan.
Suggest change to: “The Asiatic black bear (Ursus thibetanus, referred to hereafter as bears) is classified as vulnerable by the IUCN Red List due to its fragmented habitat and a clear decrease in population [2]. According to [3] about 1,000 bears live in different parts of Pakistan.” Simpler English.
Response: Changes made as suggested. Thank you!
Line 51: “Identifying these hotspots, conservation organizations can prioritize interventions, such as implementing deterrent measures or establishing buffer zones, to reduce conflicts and protect both people and wildlife [10].”
Suggest change to: “By identifying these hotspots, conservation organizations can prioritize interventions such as implementing deterrent measures or establishing buffer zones to reduce conflicts and protect both people and wildlife [10].” Simpler English.
Response: Changes made as suggested. Thank you!
Line 57: “and minimize their negative impacts by taking measures “
Suggest change to: and minimise their harm by taking measures “Simpler English.
Response: Changes made as suggested. Thank you!
Line 71: which combines spatial and temporal information from photographs of individually recognizable animals.”
Suggest change to: “which combine spatial and temporal information from photographs of individually recognisable animals ..” Two approaches are listed.
Response: Changes made as suggested. Thank you!
Line 108: “In addition to the bear, several other species live in the study area such as the snow leopard (Panthera uncia), Flare-Horned Markhor (Capra falconeri falconeri), leopard cat (Prionailurus bengalensis), yellow-throated marten (Martes flavigula), golden jackal (Canis aureus), red fox (Vulpes vulpesj, grey wolf (Canis lupus), rhesus monkey (Macaca mullata), Indian crested porcupine (Hystrix indica), and the giant Indian flying squirrel (Petaurista petaurista).”
Suggest change to: “Several other species such as the snow leopard (Ponthera uncia], flare-horned markhor (Capra falconeri falconeri), leopard cat (Prionailurus bengalensis], yellow-throated marten (Martes flavigula), golden jackal (Canis aureus), red fox (Vulpes vulpes), grey wolf (Canis lupus), rhesus monkey (Macaca mullata), Indian crested porcupine (Hystrix indica), and the giant Indian flying squirrel (Petaurista petaurista) also live in the study area.” Simpler English.
Response: We have revised this sentence as per suggestions of Reviewer 1. Thank you!
Line 119: “In each grid, we selected random plots and placed cameras at random distances from a starting point on the grid line of the selected plot. This randomization of camera stations meets the requirements of REM [39].”
Suggest change to: “In each grid, we selected random plots and placed cameras at random distances from a starting point on the grid line. Randomisation of camera stations is required for REM [39].” Simpler English.
Response: Changes made as suggested. Thank you!
Line 126: “To capture the entirety of animal movement, the camera traps were set to trigger a rapid fire of three photos upon activation, with minimal delay between each trigger. Next series of ..” Suggest change to: “To capture all animal movement, the camera traps were set to trigger three rapid fire photos, with minimal delay between each trigger. The next series of .” Simpler English.
Response: Changes made as suggested. Thank you!
Line 130: “To ensure the independence of detections, any instances of multiple detections at the same camera-trap station occurring within a 30-minute interval were excluded from the dataset [40].”
Suggest change to: “To ensure independent detections, instances of multiple detections at the same camera-trap station within 30-minutes were excluded from the dataset [40].” Simpler English. Why was 30 minutes chosen as the cut-off?
Response: Changes made as suggested. Thank you! The 30-minute interval was only used for events from which we calculated activity pattern and activity level as suggested by Ridout and Linkie, (2009) (Estimating Overlap of Daily Activity Patterns from Camera Trap Data) to exclude these photos from dataset because of concerns about lack of independence. For other parameters an event was classified as independent when an individual moved into and then out of the camera trap's field of view.
Line 147: “We considered an individual of the target species entering and exiting the field of view of the camera trap to be an independent contact.” Were there any instances of bears leaving the field of view and then re-entering it* If so, how are these anomalies handled?
Response: We counted it as a separate encounter when the animal left the field of view and then entered again after a second or minute or hour, because they are unmarked animals and we do not know if it is the same individual or not, as suggested by Palencia et al., 2022. (Random encounter model is a reliable method for estimating population density of multiple species using camera traps).
Line 147: “Second, we estimated the activity level following [41] by using the R package activity [42,43] (Fig. 2, Supplementary Materials Appendix 1). Finally, we estimated the daily range following the procedure described by [34/. The estimating method of the daily distance travelled ...” I cannot determine how the authors estimated the activity level or the day range from this information. I also cannot understand the Supplementary data provided. Much more explanation is required for this section for most readers to understand how the authors determined the data. Can the authors describe the procedure in simple terms for readers who don’t have access to the package “boot” and package “readr”in R stafisfical sofiware? Also, shouldn’t these be the “day range”rather than the “daily range”?
Response: Thank you for the feedback! The daily range has been replaced by day range as suggested. We have also included further details in the supplementary materials for readers as suggested. For further explanation, please refer to the following packages “boot, readr, and activity”. They have open access to all readers.
Canty, A. and Ripley, B., 2017. Package ‘boot’. Bootstrap Functions. CRAN R Proj.
Rowcliffe, M. and Rowcliffe, M.M., 2016. Package ‘activity’. Animal activity statistics R package version, 1.
Wickham, H., Hester, J., Francois, R., Bryan, J., Bearrows, S., Jylänki, J. and Jørgensen, M., 2024. Package ‘readr’. Read Rectangular Text Data. Available online: https://cran. r-project. org/web/packages/readr/readr. pdf (accessed on 23 August 2023).
Line 187: “The day range of the bear population averaged 21.36 + 9.64 km/day (mean 1 SE). With such a large standard error there seems no logical reason to provide an accuracy to 2 decimal places. Even one decimal place is suspect.
Response: Reviewer 1 had concern regarding the potential underestimation of uncertainty in our population density estimate. To address that, we have implemented a two-step approach to ensure a comprehensive assessment of uncertainty. First, we employed bootstrapping with 10,000 iterations to estimate the variance associated with all parameters involved in the Random Encounter Model, including the encounter rate, day range, and the radius and angle of detection. This bootstrapping approach allowed us to capture the variability in each parameter, providing a robust estimation of their respective variances. Second, we used the delta method to calculate the total variance in density estimates, considering the variances of all parameters obtained from the bootstrapping step. By integrating the variances of all relevant parameters, the delta method provides a more accurate and conservative estimation of the total variance in density estimates. We have added a detailed description of this approach and its results to the manuscript in (Supplementary materials appendix 1 and 2), demonstrating its effectiveness in providing a more realistic assessment of uncertainty.
The result from this study and those published from other bear density studies vary greatly. The authors suggest these great differences are likely due to the unusually high density of bears in the area at this time of the year due to the greater food availability (acorns and cultivated fruits). But it could also be due to the use of different approaches (hair sample genetic analyses, questionnaire surveys compared with camera-trap and REM methods). Until the accuracy of these different approaches have been determined we will not know what the main reasons are for the differences and thus how valuable the findings of the paper are. The authors also state that to improve bear elevations and to investigate how depletion or abundance of acorns or other natural food resources are associated with human-bear conflicts in the study area. As none of this information is included in the paper it seems the true value of this article cannot be known until these additional investigations have also been undertaken. So, the publication of this study seems premature.
Response: Thank you for your feedback! To enhance our study, we have not only estimated the population density using the Random Encounter Model but have also added a hotspot analysis of bear encounter rates using kernel density estimation in ArcGIS 10.8. Further, we conducted an exploratory regression analysis in ArcGIS 10.8, incorporating a variety of ecological factors, to deepen the understanding of Asiatic black bear density in this understudied area. These additions aim to provide a solid foundation for the conservation and management of the threatened Asiatic black bear and fill a significant gap in literature.
Reviewer 3 Report
Comments and Suggestions for Authors
The authors used camera traps data to analyze the density of Black bears in the Hindu Raj Mountains, Pakistan. Considering one of the least studied species from previously not studied area, the research is important. However, the manuscript suffers from many limitations.
The results presented in the manuscript do not carry the title. The overall manuscript neds to be improved.
The number of camera traps used and trap nights are too small.
The most important limitation is, despite the title demanding it, there is nothing presented in the results section about the seasonal variation in the population density of the black bear between autumn and winter.
The authors have been stating about the hotspots of habitat used by the bears, however they have done no hotspot analysis. It should be added in the manuscript.
Line 95-96: ..... the study area remains undocumented: This alone is not the research question. The authors need to justify the need of this study in Hindu Raj mountain.
The methods section should state clearly about when to when, for how many days and total how many hours were the cameras deployed in the field.
Line 174: the mean is smaller than the coefficient of variation. Is it true? Please recheck.
Table 1 is not necessary as it just contains one row. The information can be presented in one/two lines of text.
Overall, the manuscript has limited data and very few analyses. The results do not seem to be enough for the full article.
There are inconsistencies in the writeup and the ways of citing literature.
Please find the annotated PDF.

Author Response
The authors used camera traps data to analyze the density of Black bears in the Hindu Raj Mountains, Pakistan. Considering one of the least studied species from previously not studied area, the research is important. However, the manuscript suffers from many limitations.
The results presented in the manuscript do not carry the title. The overall manuscript needs to be improved.
Response: We have divided the Results into three sub-headings, Bears density, Hotspots of bears encounter rates and Results of Exploratory regression analysis.
The number of camera traps used, and trap nights are too small.
Response: Our camera trapping survey resulted in 66 independent detections of bears (minimum of 30 minutes interval between each two photos were considered for independent detections). The random encounter model requires that cameras should be deployed for as long as is necessary to obtain a minimum of 10 photographs but preferably at least 20 (Rowcliffe et al., 2008). In Rovero et al. (2013) there is a personal communication from Rowcliffe that suggests the minimum number of photographs should be at least 50.
The most important limitation is, despite the title demanding it, there is nothing presented in the results section about the seasonal variation in the population density of the black bear between autumn and winter.
Response: The number of independent photographs as suggested by Rowcliffe in a personal communication in Rovero et al., (2013), that there should be at least 50 photographs. We did not have enough independent detections for each season, so the density was calculated for both seasons combined.
The authors have been stating about the hotspots of habitat used by the bears; however they have done no hotspot analysis. It should be added in the manuscript.
Response: Hotspot analysis added as suggested. Thank you!
Line 95-96: .... the study area remains undocumented: This alone is not the research question. The authors need to justify the need of this study in Hindu Raj Mountain.
Response: Research question (In this study, we investigated that how the seasonal migration of Asiatic black bears to lower elevations during autumn and winter influence their population density, spatial distribution, and interactions with human settlements in the Hindu Raj Mountains, and what are the implications for their conservation and management in this unique ecosystem) added as suggested. Thank you!
The methods section should state clearly about when to when, for how many days and total how many hours were the cameras deployed in the field.
Response: Thes suggested information has been added in methods section. Thank you!
Line 174: the mean is smaller than the coefficient of variation. Is it true? Please recheck.
Response: It was variance associated with mean, not mean. But, as per the suggestions and comments of Reviewer 1, we have revised the density estimation calculations. (Supplementary materials appendix 1 and 2). Thank you for the feedback!
Table 1 is not necessary as it just contains one row. The information can be presented in one/two lines of text.
Response: While we appreciate the suggestion, we believe retaining the table format for presenting the main results, including encounter rate, speed, detection radius, angle, and density along with its standard error, provides clarity and ease of reference for readers, despite its brevity.
Overall, the manuscript has limited data and very few analyses. The results do not seem to be enough for the full article.
Response: Thank you for the feedback! We have incorporated hotspot analysis as suggested, and there was the opportunity to investigate the relationship of bears encounter rates with forest cover, elevation, distance to built in areas (villages), slope, aspect, and roughness. The details incorporated in the revised article are given below.
Hotspot analysis
To identify bear encounter hotspots within the study area, we employed methods based on Kernel Density Estimation (KDE). KDE is a widely used technique for visualizing and analyzing spatial data, with the objective of understanding and potentially predicting patterns of events. Utilizing the "Kernel Density for Point Features" tool in ArcGIS 10.8, we calculated the point feature density around each output raster cell. The algorithm of this tool fits a smoothly curved surface over each point, with the surface value being highest at the point location and decreasing as the distance from the point increases, eventually reaching zero at the search radius (bandwidth). This approach is particularly effective in identifying hotspots because it makes a series of density estimates over a grid that covers the entire point pattern. By applying KDE to our bear encounter data, we were able to identify areas of high bear activity visually and quantitatively, which are crucial for understanding spatial patterns and informing conservation efforts.
Exploratory regression analysis
We employed the Exploratory Regression tool in ArcGIS 10.8 to investigate the relationships between bear encounter rates and a set of environmental variables, including Euclidean distance to built-in areas, altitude, forest cover, aspect, slope, and roughness (further details in the Supplementary Materials, Appendixes 5).To investigate the relationship between bear encounter rate and environmental variables, we utilized raster data for altitude, aspect, slope, and roughness, which were downloaded from Open Topography. Forest cover data were obtained from the Global Forest Change project. The Euclidean distance to built-in areas was determined using a raster developed from Esri's Land Use Land Cover data. These environmental variables were selected based on their potential influence on bear habitat preferences and movement patterns. The analysis systematically evaluated all possible combinations of the input explanatory variables, aiming to identify Ordinary Least Squares (OLS) models that best explained the dependent variable, encounter rates, within user-specified criteria. Key parameters set for the analysis included a maximum number of explanatory variables of 6, a minimum acceptable adjusted R-squared of 0.3, a maximum coefficient p-value cutoff of 0.05, a maximum Variance Inflation Factor (VIF) value cutoff of 7.5, and minimum acceptable p-values for the Jarque-Bera and spatial autocorrelation tests of 0.1. The results from this analysis provided insights into the ecological dynamics influencing bear distribution, serving as a robust framework for understanding habitat preferences and guiding conservation efforts.
There are inconsistencies in the writeup and the ways of citing literature.
Response: We have removed the inconsistencies in writeup. Thank you!
Round 2
Reviewer 1 Report
Comments and Suggestions for Authors
The authors did a great job in implementing the previous comments.
I have only a few small comments:
Line 147: Please rearrange the sentence:"we excluded multiple detections at the same camera-trap station within 30 minutes from the dataset."
Line 195: Please specify that effective detection radius and angle were computed as the mean.
Line 243: Please write "Bear density" instead of "Bears density".
Line 233: I would suggest to mention how many models you ended up with in total to better interpret the percentages in the results.
Author Response
Reviewer 1
The authors did a great job in implementing the previous comments.
R: Thank you very much for your congratulations.
I have only a few small comments:
Line 147: Please rearrange the sentence:"we excluded multiple detections at the same camera-trap station within 30 minutes from the dataset."
R: Done. Now you read: “To ensure independent detections for calculating activity level, we deleted from our dataset multiple detections obtained from the same camera-trap station within 30-minutes”.
Line 195: Please specify that effective detection radius and angle were computed as the mean.
R: Done.
Line 243: Please write "Bear density" instead of "Bears density".
R: Done.
Reviewer 2 Report
Comments and Suggestions for Authors
The changes made to the report have significantly improved it. I have no further comments as my experience with the analytical techniques employed is limited. All I can say is I can see no current errors that need addressing.
Author Response
Reviewer 2
The changes made to the report have significantly improved it. I have no further comments as my experience with the analytical techniques employed is limited. All I can say is I can see no current errors that need addressing.
R: Thank you very much for your valuable contribution to the review process.